Enhancing small-scale acetification processes using adsorbed Acetobacter pasteurianus UMCC 2951 on κ-carrageenan-coated luffa sponge

Sriphochanart Wiramsri 1
Krusong Warawut 1
Samuela Nialmas 1
Somboon Pichayada 1
Sirisomboon Panmanas 2
Onmankhong Jiraporn 2
Pornpukdeewattana Soisuda 1
Charoenrat Theppanya thepcharoen4@gmail.com 3
1 Division of Fermentation Technology, School of Food Industry, King Mongkut’s Institute of Technology Ladkrabang , Bangkok , Thailand
2 Department of Agricultural Engineering, School of Engineering, King Mongkut’s Institute of Technology Ladkrabang , Bangkok , Thailand
3 Department of Biotechnology, Faculty of Science and Technology, Thammasat University , Pathum Thani , Thailand
Okpala Charles
Electronic publication date: 2024 Jun 28
Publication date: 2024
Volume: 12
Electronic Location ID: e17650
Received 2024 Jan 19; Accepted 2024 Jun 7
Copyright: ©2024 Sriphochanart et al.
Copyright year: 2024
Copyright holder: Sriphochanart et al.
License: This is an open access article distributed under the terms of the Creative Commons Attribution License, which permits unrestricted use, distribution, reproduction and adaptation in any medium and for any purpose provided that it is properly attributed. For attribution, the original author(s), title, publication source (PeerJ) and either DOI or URL of the article must be cited.
License URL: https://creativecommons.org/licenses/by/4.0/

Keywords: Acetification, Acetobacter pasteurianus, Adsorption, κ-carrageenan, Luffa sponge, Acetic acid bacteria, Bacterial cellulose

Funding: The King Mongkut’s Institute of Technology Ladkrabang Research fund KREF116101 KRF046008 This work was supported by the King Mongkut’s Institute of Technology Ladkrabang Research fund (KREF116101 and KRF046008). The funders had no role in study design, data collection and analysis, decision to publish, or preparation of the manuscript.

==============================
Background

This study explored the utilization of luffa sponge (LS) in enhancing acetification processes. LS is known for having high porosity and specific surface area, and can provide a novel means of supporting the growth of acetic acid bacteria (AAB) to improve biomass yield and acetification rate, and thereby promote more efficient and sustainable vinegar production. Moreover, the promising potential of LS and luffa sponge coated with κ-carrageenan (LSK) means they may represent effective alternatives for the co-production of industrially valuable bioproducts, for example bacterial cellulose (BC) and acetic acid.

Methods

LS and LSK were employed as adsorbents for Acetobacter pasteurianus UMCC 2951 in a submerged semi-continuous acetification process. Experiments were conducted under reciprocal shaking at 1 Hz and a temperature of 32 °C. The performance of the two systems (LS-AAB and LSK-AAB respectively) was evaluated based on cell dry weight (CDW), acetification rate, and BC biofilm formation.

Results

The use of LS significantly increased the biomass yield during acetification, achieving a CDW of 3.34 mg/L versus the 0.91 mg/L obtained with planktonic cells. Coating LS with κ-carrageenan further enhanced yield, with a CDW of 4.45 mg/L. Acetification rates were also higher in the LSK-AAB system, reaching 3.33 ± 0.05 g/L d as opposed to 2.45 ± 0.05 g/L d for LS-AAB and 1.13 ± 0.05 g/L d for planktonic cells. Additionally, BC biofilm formation during the second operational cycle was more pronounced in the LSK-AAB system (37.0 ± 3.0 mg/L, as opposed to 25.0 ± 2.0 mg/L in LS-AAB).

Conclusions

This study demonstrates that LS significantly improves the efficiency of the acetification process, particularly when enhanced with κ-carrageenan. The increased biomass yield, accelerated acetification, and enhanced BC biofilm formation highlight the potential of the LS-AAB system, and especially the LSK-AAB variant, in sustainable and effective vinegar production. These systems offer a promising approach for small-scale, semi-continuous acetification processes that aligns with eco-friendly practices and caters to specialized market needs. Finally, this innovative method facilitates the dual production of acetic acid and bacterial cellulose, with potential applications in biotechnological fields.

Introduction

Luffa sponge (LS) is derived from the fibrous mature fruit of the tropical plant Luffa cylindrica, a member of the Cucurbitaceae family. The versatile fibers serve not only as cleaning tools but also in biotechnological applications where their biodegradability and eco-friendliness are advantageous (Ha et al., 2023). Studies have highlighted LS as an adsorption material for various microorganisms on account of its high porosity and stable physical properties (Krusong et al., 2021; Rahman et al., 2021; Alasali et al., 2023). Importantly, the fibrous structure can support the growth of acetic acid bacteria (AAB), facilitating the efficient luffa sponge-acetic acid bacteria (LS-AAB) vinegar production system (Krusong et al., 2020).

AAB are members of the Acetobacteriaceae family in the Alphaproteobacteria class, and are recognized for their capacity to oxidize ethanol, converting wine to vinegar (Qiu, Zhang & Hong, 2021). While many genera of AAB exist, only Acetobacter, Gluconacetobacter, and Komagataeibacter are considered ideal for industrial vinegar production; these three genera tolerate extreme conditions and produce high amounts of acetic acid (Román-Camacho et al., 2023). Historically, vinegar production has been dominated by two main processes: the traditional surface culture acetification and the faster submerged semi-continuous process (Silva et al., 2024). In the traditional method, AAB grow on the substrate surface, forming a “mother of vinegar”—a jelly-like mass often used as a starter culture (Neffe-Skocińska et al., 2023). However, among semi-continuous methods, the quick acetification process (QAP) has become preferred for industrial applications (Gullo & Giudici, 2008; Krusong et al., 2020) and is economically favorable for small-scale production. In this process, the AAB are adsorbed on materials such as corncobs, oak wood chips (Hutchinson et al., 2019), delignified cellulosic material (Plioni et al., 2021), beech shavings, zeolite, and high-pressure polyethylene (Panasyuk et al., 2021), and, notably, LS (Krusong et al., 2020).

Many microorganisms have the capability to form biofilms, structured communities attached to surfaces (Hayta et al., 2021). These biofilms consist mainly of an extracellular polymeric substance (EPS) matrix, which includes polysaccharides, polynucleotides, and proteins (Flemming et al., 2023). Biofilm formation involves direct cell–cell contact and adherence to the solid–liquid interface, and is crucial in maintaining various cellular functions (Sanchez-Vizuete et al., 2022). The polysaccharides within the EPS matrix may be synthesized extracellularly, or intracellularly and then secreted into the environment (Wang et al., 2022). Santos et al. (2024) discovered that many bacteria can alternate between biofilm and planktonic states. Of the various biofilm types, bacterial cellulose (BC) is notable for its unique fibrous structure.

This study focused on small-scale vinegar production, specifically with a reciprocal shaking acetification process. Krusong & Tantratian (2014) initially proposed this process as an economical and eco-friendly alternative in which the AAB can be adsorbed to organic materials. However, they identified formation of BC biofilm as a major challenge in the method, hindering acetification and reducing yield. Notably, such films are typically seen in static acetification environments. Thus, this work aimed to enhance acetification efficiency by applying reciprocal shaking at a speed of 1 Hz; it further investigated the effect of applying a κ-carrageenan coating to the LS surface on acetic acid production, BC formation, productivity, and yield. Finally, the study explored the potential advantages of co-producing acetic acid and BC, which may offer unique benefits for this process.

Materials & Methods

Materials, sample preparation, and acetification equipment

Materials: LS gourd samples, approximately six cm in diameter and 20–25 cm in length, were obtained from a local market in Ladkrabang, Bangkok, Thailand. Pineapple wine (100.0 ± 0.4 g/L ethanol, 0.9 ± 0.1 g/L acidity) and pineapple vinegar (80.0 ± 0.3 g/L acidity) were supplied by the Laboratory of Fermentation Technology, School of Food Industry, King Mongkut’s Institute of Technology Ladkrabang, Bangkok, Thailand. The fermentation medium was prepared using both pineapple wine and pineapple vinegar to initiate the acetification process; it was also supplemented with several nutrients, including glucose, yeast extract, malt extract, diammonium hydrogen phosphate ((NH4)2HPO4), and magnesium sulfate (MgSO4.7H2O), which were sourced from Merck KGaA, Darmstadt, Germany. Finally, κ-carrageenan was procured from Foodmate Co., Ltd., Shanghai, China.

LS preparation: For acetification, LS gourds were sectioned into cylindrical pieces six cm in diameter and 2.5 cm thick. These pieces were first cleaned by washing in tap water for 10 min, soaking in 5% v/v acetic acid (derived from pineapple vinegar) for an hour, and finally undergoing two 10-minute tap water rinses. Post-cleaning, the pieces were air-dried in a Biological Safety Cabinet (AC24S8NS, ESCO, St. Louis, MO, USA) for 30 min, weighed, packaged in plastic, and sterilized by heating in two 30-minute sessions at 121 °C. Sterile LS pieces were stored at an ambient temperature of 32 ± 2 °C.

Acetification equipment: The acetification setup was as demonstrated by Krusong & Tantratian (2014), designed with the aim of enhancing oxygenation in the QAP for small-scale vinegar production. The setup consisted of a 60-liter plastic tank (0.5 m × 0.4 m × 0.3 m) mounted on a reciprocating steel plate (1.00 m × 0.50 m × 0.025 m). An electric motor (Hitachi 220v, 1 phase, 4 pole, split phase motor, 14 HP, Samut Prakan, Thailand) connected to a belt and pulley system (Chenggang Electrical Engineering Co., Ltd., Bangkok, Thailand) operated a crank to create a reciprocating motion with a 0.25 m amplitude. The shaking frequency was controlled at 1 Hz using a FRN0.4G1S-4A inverter from Fuji Electric Inc., Hunghom, Kowloon, Hong Kong.

LS properties analysis: Twelve random LS samples were characterized in terms of properties such as weight, volume, true density, bulk density, porosity, and specific surface area. Determinations of weight, volume, true density, and bulk density were adapted from Ogunsina, Adegbenjo & Opeyemi (2010). True density and bulk density were calculated using Eqs. (1) and (2) as follows: (1) Dtg/cm3=WdgVtcm3

(2) Dbg/cm3=WdgVbcm3

where Dt represents true density, Db represents bulk density, Wd is the dry weight, Vt is the true volume, and Vb is the bulk volume (including pores). Vt was measured by the water displacement method, while Vb was calculated as the geometric volume of the cylindrical piece.

Porosity, quantified as the ratio of pore volume to total volume (including pores), was expressed as a percentage. Specific surface area (m2/g) refers to the surface area per unit mass. These parameters were measured using a Mercury Intrusion Porosimeter: AutoPore V (Micromeritics, Norcross, GA, USA) at the Center of Scientific Equipment for Advanced Research, Thammasart University, Pathum Thani, Thailand.

Microorganism

The microorganism utilized in this study was the acid-tolerant Acetobacter pasteurianus strain UMCC 2951. Originally isolated from ripe pineapples, this strain has been maintained in culture for over a decade (Pothimon et al., 2020). The starter culture was prepared following the protocol of Krusong et al. (2020) using a complex medium comprised of 50 g/L glucose, 5 g/L yeast extract, 0.2 g/L MgSO4.7H2O, and 0.5 g/L (NH4)2HPO4. The culture was incubated at 32 ± 2 °C with an aeration rate of 1 L/L minute (1 vvm) for a period of 7 days.

Reciprocal shaking semi-continuous acetification process

The acetification process employed in this study was based on the high initial acetic acid concentration approach by Krusong, Yaiyen & Pornpukdeewatana (2015), with the total solute concentration standardized to 80 g/L. The concentrations of ethanol (from pineapple wine) and acetic acid (from pineapple vinegar) were respectively adjusted to 35 ± 1 g/L and 45 ± 1 g/L, as per Eq. (3): (3) NtotalVtotal=N1V1+N2V2

where N 1 and N 2 respectively represent the concentrations of ethanol in pineapple wine (g/L) and acetic acid in pineapple vinegar (g/L); V1 and V2 the required volumes of pineapple wine (L) and pineapple vinegar (L); Ntotal represents the target concentration in the fermentation medium; and Vtotal is the total volume of medium required.

The fermentation medium was enriched with the same nutrients as the medium used in starter preparation, following Krusong et al. (2020). The acetic acid concentration in the prepared fermentation medium was 45 g/L, sufficient to exert an inhibitory effect on contaminants; therefore, sterilization was not required. To carry out fermentation, 24 L of prepared medium was placed in the 60-liter plastic tank (40% of the tank’s volume), inoculated with A. pasteurianus UMCC 2951 starter culture (initial concentration 5% v/v), and acetification allowed to proceed for 30 days under reciprocal shaking at 1 Hz and temperature 32 ± 2 °C.

During the fermentation process, once the ethanol concentration decreased to 5 g/L or less, about 9.6 L (40% of the initial volume) of the fermented broth was harvested as vinegar, as recommended by Krusong et al. (2020). To initiate the next acid production cycle, the same volume of fresh medium was added, consistent with protocols from Ndoye et al. (2007) and Krusong & Tantratian (2014). Ethanol content, acidity, and cell dry weight (CDW) were measured daily, and BC biofilm formation was monitored in each cycle. When biofilm formation occurred, BC was quantified at cycle end, and selected LS samples were examined by scanning electron microscopy (SEM).

Improved acetification process using acetic acid bacteria immobilized on luffa sponge

To enhance the acetification process, sterile LS pieces were introduced into a 60-liter plastic tank to achieve a sedimented volume of 24 L (40% of the tank’s volume). Subsequently, 24 L of the starting medium was added and the tank was positioned on a reciprocating shaker. Setup, sampling, and analysis procedures were as described in the “Acetification Equipment” section above. This process is henceforth termed the LS-AAB process.

Acetification process using luffa sponge coated with κ-carrageenan for acetic acid bacteria immobilization

κ-carrageenan was utilized to improve gel stability under the acid condition of the acetification process (Gustaw & Mleko, 2003). The coating solution was prepared by dissolving κ-carrageenan in normal saline (8.5 g/L NaCl) to obtain a concentration as high as 3% (w/v) and then heating the solution to 70−80 °C. To apply the gel solution to LS pieces, the pieces were immersed for 15 s, then air-dried in a Biological Safety Cabinet (AC24S8NS, ESCO, St. Louis, MO, USA) for 30 min. The experimental setup and conditions for the acetification process were the same as for uncoated LS. This process is henceforth termed the luffa sponge coated with κ-carrageenan-acetic acid bacteria (LSK-AAB) process.

SEM observation of the adsorbed materials during the acetification process

Cell adsorption and BC biofilm formation on LS samples was determined by means of SEM for both the standard LS-AAB process and the modified LSK-AAB process. All samples were gold-coated for 45 s by a rotary pumped coater (Q150R ES Plus; Quorum, East Sussex, United Kingdom). SEM was performed using a SEM-EDS (EVO MA10, ZEISS, Germany) operated at an accelerating voltage of 20 kV. Images were taken to determine the microbial interactions and biofilm formation on LS and LSK surfaces.

Analytical methods

Vinegar acidity, expressed as % w/v, was measured according to the method of Krusong et al. (2020), namely acid–base titration using 0.1 mol/L NaOH with phenolphthalein as indicator. Ethanol content was quantified using a gas chromatograph (GC-FID; Agilent Technologies, Palo Alto, CA, USA) equipped with a headspace injector and a flame ionization detector (FID). Samples were placed in sealed headspace vials containing a known quantity of isopropanol as an internal standard, then heated to 80 °C for an equilibration time of 15 min. The injector needle temperature was maintained at 85 °C, the injector itself was kept at 150 °C, and the injection volume was 200 µL. Separation was achieved using a J&W DB-WAXetr column (0.53 mm × 30 m, 2 µm film; Agilent Technologies, Palo Alto, CA, USA). The temperature of the FID detector was set at 250 °C. Helium served as the carrier gas and was supplied at a flow rate of seven mL/min. The oven temperature was initially held at 40 °C for 10 min, then ramped up at a rate of 25 °C/min to a final temperature of 240 °C, which was maintained for 1 min. Each sample had a total run time of 20 min. This analytical method was adapted from Liu et al. (2019).

Cell dry weight (CDW) in the fermentation broth was determined from the absorbance measured at 660 nm (OD660) using a spectrophotometer (GENESYS 10VIS; Thermo Fisher Scientific, Waltham, MA, USA) on the basis of a standard curve. Samples were diluted to ensure an OD660 between 0.3 and 0.8. Adhered A. pasteurianus UMCC 2951 cells on LS-AAB or LSK-AAB were first detached by adding 50 mM sodium citrate solution and then agitating at 1.5 Hz for 1 h at 32 ± 2 °C (Phisalaphong et al., 2007). The obtained cell suspension was subsequently assessed for OD600 and quantified for adsorbed CDW.

For BC formation analysis, residual bacterial cells and medium components were first removed as per Al-Shamary & Al-Darwash (2013); this involved washing with distilled water, treatment with 1 M NaOH for 15 min at 100 °C, rinsing until neutral pH, and finally drying at 100 °C to a constant weight. Subsequent analysis followed the method of Jung, Park & Chang (2005).

Dissolved oxygen (ppm) was monitored using a HI9146 Microprocessor Dissolved Oxygen Meter (HANNA Instruments Inc., Cluj Napoca, Romania) (Krusong & Tantratian, 2014).

Statistical analysis

In this study, all experiments were performed in triplicate. The results are presented as mean values plus or minus the corresponding standard deviation. Significant differences among treatments were identified by means of one-way analysis of variance (ANOVA) with a significance level of p = 0.05. Pairwise differences in means were likewise evaluated using Tukey’s Honestly Significant Difference (HSD) test with a significance level of p ≤ 0.05. Statistical analyses were performed using IBM SPSS version 24.0.

Results

Properties of LS used in the semi-continuous acetification process

LS is recognized as a versatile biotechnological tool and its distinctive properties have attracted considerable attention across multiple sectors (Akinyemi & Dai, 2022). In particular, LS is notable for its porous structure, substantial degree of lignification (Li, Wang & Zhang, 2024), and significant chemical and physical stability (Shen et al., 2012); (Moreno-Anguiano et al., 2021). Figure 1 illustrates the dried LS samples employed in this study, which were prepared as short cylindrical cross-sections of about six cm in diameter and 2.5 cm in height. These samples present a shuttle-like, fibrous layer structure (Fig. 1A) comprised of outer, inter, middle, and inner sections. Significantly, the fiber bundles in the middle and inner sections were thicker compared to those in the outer and inter sections (Figs. 1B–1D). Our measurements of fiber dimensional characteristics are in agreement with previous reports, which include fiber cell lengths of 1,050–1,070 µm, diameters of 17–28 µm, lumen diameters of 12–24 µm, and wall thicknesses of 3–8 µm (Chen et al., 2017).

Figure 1 Representative images of LS showing fiber characteristics in different sections: (A) the outer surface and inner part; (B) the middle part in longitudinal section; (C) the middle, inter, and inner parts in horizontal section; (D) fibers from the outer and inner surfaces in longitudinal section.

This study assessed the physical properties of L. cylindrica LS and its associated suitability as an adsorbent for A. pasteurianus UMCC 2951; the results are presented in Table 1. Importantly, the obtained true density of 0.82 ± 0.02 g/cm3 surpasses the 0.57 ± 0.012 g/cm3 previously reported for L. aegyptiaca (Ogunsina, Adegbenjo & Opeyemi, 2010). Meanwhile, the low bulk density of (2.48 ± 0.05) × 10−2 g/cm3, alongside significant porosity (54.55 ± 3.44%) and specific surface area (4.24 ± 0.17 m2/g), suggests a larger void fraction. This advantageous structure is key to both efficient adsorption of A. pasteurianus UMCC 2951 and facilitation of oxygen transfer, which are required for effective acetification.

Table 1 Values of LS parameters important for its use as an adsorbent in semi-continuous acetification.

Parameter	Value	
Weight (g/piece)	1.68 ± 0.06	
True density (g/cm3)	0.82 ± 0.02	
Bulk density (g/cm3)	(2.48 ± 0.05) × 10−2	
Porosity (%)	54.55 ± 3.44	
Specific surface area (m2/g)	4.24 ± 0.17	
Notes.

Values are the average ± standard deviation (SD) measured from twelve pieces of LS.

Semi-continuous acetification using planktonic cells and reciprocal shaking

Consistent with the findings of Krusong & Tantratian (2014), acetification results in the present study supported the reciprocating shaker as vital for adequate oxygenation, a key factor in small-scale acetification. Over the course of the 30-day experiment, the planktonic system facilitated just two acetification cycles. The startup phase was notably prolonged, extending for 12 days as illustrated in Fig. 2A.

Figure 2 Acetification by A.  pasteurianus UMCC 2951 in a reciprocating shaking system at 32 ± 2 °C: (A) planktonic cells; (B) cells adsorbed on luffa sponge (LS); and (C) cells adsorbed on luffa sponge coated with κ-carrageenan (LSK).

Table 2 presents the acetification efficiency, biomass accumulation, and BC formation obtained for planktonic cells across various cell states and LS modifications. Overall, the average CDW was 0.91 mg/L. A critical aspect of this study involved testing the resilience of A. pasteurianus UMCC 2951 under high acidity conditions (initial acetic acid concentration 45 g/L). These cells exhibited strong tolerance of acidity, achieving an average acetification rate of 1.13 ± 0.05 g/L d. Notably, BC and polysaccharide gel did not form during acetification with planktonic cells. Understanding this aspect is essential for a comprehensive grasp of acetification dynamics.

The acetification process typically involves two phases: a startup or adaptation phase, in which the AAB adapt to their environment, followed by an operational phase focused on acetic acid production. To emulate the semi-continuous process common in vinegar production, 40% of the initial culture volume (approximately 9.6 liters) was discharged and replenished at the end of each cycle. This approach, referred to as “semi-continuous acetification,” aligns with the “successive discontinuous cycles of acetification” described by Kocher & Dhillon (2013). Kumar & Kocher (2017)) similarly employed semi-continuous fermentation for sugarcane vinegar production by Acetobacter aceti.

According to Merli et al. (2021), the growth of AAB typically decelerates at acetic acid concentrations above 40 g/L; however, certain strains, such as A. pasteurianus, exhibit resistance to these elevated concentrations, a tolerance essential for efficient ethanol oxidation (Han et al., 2020; Gao et al., 2021). The present study used a high initial acetic acid concentration of 45 g/L, and the obtained acetification rate of 1.13 ± 0.05 g/L d corroborates the acidity resistance of A. pasteurianus; this rate is comparable to values reported in similar studies (Hidalgo et al., 2010; Qi et al., 2013).

Improved acetification process using cells immobilized on luffa sponge

When AAB were immobilized on LS (the LS-AAB system), the startup phase was notably shortened, lasting only 9 days. Furthermore, the LS-AAB system completed four acetification cycles over the 30-day experimental period, marking a significant increase in productivity versus planktonic cells (Fig. 2B). The success of this system was primarily due to enhanced accumulation of A. pasteurianus UMCC 2951 biomass, as reflected in the combined CDW of 3.34 mg/L for both adsorbed and planktonic cells (Table 2). This increase in biomass directly correlated with an improved acetification rate of 2.45 ± 0.04 g/L d. The high surface area of LS was pivotal in enhancing oxygen transfer, a crucial factor for effective acetification. However, in later cycles, formation of BC and polysaccharide gel blocked oxygen diffusion and consequently impeded acetification efficiency.

The cell wall of AAB is likely to influence bacterial biomass accumulation on LS. A. pasteurianus UMCC 2951 is a gram-negative bacterium; such bacteria are characterized by cell surfaces rich in negatively charged components such as lipopolysaccharides (Marchetti et al., 2021) and therefore typically possess a negative surface charge at physiological pH levels. Meanwhile, the charge properties of LS suggest that it carries a positive charge in solutions with a pH below 4.5, and a negative charge in more alkaline environments. Consequently, under acidic conditions, the positive charge of A. pasteurianus UMCC 2951 cells can enable the formation of strong ionic bonds with the LS surface, leading to substantial biomass accumulation as adsorbed cells (Table 2).

At the same time, the configuration of LS also supported effective air circulation within the reciprocating shaking system, creating a highly oxygenated environment conducive to AAB growth and proliferation. In conjunction with the high biomass of both adsorbed cells (2.40 mg/L) and planktonic cells (0.94 mg/L), this led to an impressive average acetification rate of 2.45 ± 0.04 g/L d, observed over 30 days at 32 ± 2 °C. Importantly, this culminated in a marked improvement in acetic acid production when compared to systems utilizing only planktonic cells.

Table 2 Acetification process outcomes obtained using planktonic cells, LS-AAB, and LSK-AAB.1

	Planktonic cells	LS-AAB	LSK-AAB	p -value 2	
Startup phase (d)	12	9	7		
Number of acetification cycles	2	4	5		
Acetification period (d)	8	5	4.2		
Final acid concentration (g/L)	54.00 ± 0.30c	57.25 ± 0.40b	59.00 ± 0.40a	<0.001	
Acetic acid produced (g/L)	9.00 ± 0.30c	12.25 ± 0.40b	14.00 ± 0.40a	<0.001	
Acetification rate (g/L d)	1.13 ± 0.05c	2.45 ± 0.04b	3.33 ± 0.05a	<0.001	
CDW (mg/L)					
- Planktonic cells in medium	0.91 ± 0.002c	0.94 ± 0.001b	0.95 ± 0.002a	<0.001	
- Absorbed cells	No LS added	2.40 ± 0.04b	3.50 ± 0.07a	<0.001	
Cycle at which BC formed	BC not formed	Cycle 3–4	Cycle 3–5		
BC produced (mg/L)	BC not formed	25.00 ± 2.00b	37.00 ± 3.00a	0.004	
Dissolved oxygen (ppm)	4.12 ± 0.20a	2.98 ± 0.20b	2.53 ± 0.10c	<0.001	
Notes.

1 Values are presented as mean ± standard deviation. Different letters within a row indicate significant difference (p ≤ 0.05) as determined by Tukey’s test.

2 For p-values greater than or equal to 0.001, precise values are shown. Values less than 0.001 are reported as p < 0.001.

It is also notable that acetic acid production with this system peaked in the second cycle; cycles 3 and 4 exhibited a notable decline in production. This decreased rate was attributed to the formation of BC microfibrils and polysaccharide gel, with 25.0 ± 2.0 mg/L BC produced on average in cycles 3 and 4. These structures act as barriers to oxygen diffusion, consequently diminishing AAB oxidative activity and reducing acetic acid production (Tantratian et al., 2005). This result highlights the difficulty in maintaining an environment conducive to acetification.

Yun, Kim & Lee (2019) suggested that BC production by A. pasteurianus serves as a defense mechanism against elevated concentrations of acetic acid. This phenomenon has also been observed in other strains of A. pasteurianus, including RSV-4 (Kumar et al., 2021) and MGC-N8819 (Nie et al., 2022). Several studies have also reported A. pasteurianus to form a protective biofilm of specific pellicle polysaccharides during acetification, which prevent acetic acid from penetrating into the cytoplasm (Samyn et al., 2023). This natural defensive response highlights the intricate interplay between biological and chemical factors over the course of the acetification process.

Enhancement of acetification rate with κ-carrageenan-coated luffa sponge

When using κ-carrageenan-coated luffa sponge, significant increases in acetic acid production and acetification rate were observed; specifically, the LSK system achieved the highest mean acetification rate of 3.33 ± 0.05 g/L d, significantly different (p < 0.001) from the 2.45 ± 0.04 g/L d of LS-AAB and 1.13 ± 0.05 g/L d of planktonic cells. In addition, the κ-carrageenan coating reduced the startup phase to just 7 days and enabled five operational cycles to be completed within the 30-day experiment period, as depicted in Fig. 2C.

LSK demonstrated greater CDW of absorbed cells, specifically 3.50 ± 0.07 mg/L versus the 2.40 ± 0.04 mg/L obtained with LS-AAB (p < 0.001). This greater biomass yield could contribute to the greater metabolic rate and more efficient vinegar production. However, BC production during later cycles was also significantly higher in the LSK-AAB system, at 37.00 ± 3.00 mg/L compared to 25.00 ± 2.00 mg/L for LS-AAB (p = 0.004). This suggests that the κ-carrageenan coating not only enhances cell attachment and growth but also promotes biofilm formation. While this characteristic is beneficial for certain biotechnological applications, it may pose challenges to oxygen transfer and mass efficiency during acetification, as discussed above. Nonetheless, LSK-AAB also achieved a significant increase in acetic acid production, reaching a peak concentration of 14.0 ± 0.4 g/L, the highest observed in this study. The gelling properties of κ-carrageenan were likely critical to this improvement (Zia et al., 2017). κ-carrageenan itself naturally shows cationic properties (Wang et al., 2023), especially in the presence of cations; these properties facilitate a robust interaction with the anionic sites on bacterial cell surfaces and enhance cell adsorption onto the support matrix, thereby improving biofilm stability and efficacy.

In terms of operational efficiency, the LSK-AAB system was exceptional, with the average cycle duration being only 4 days over five cycles. The production of acetic acid in each cycle varied from 56 to 65 g/L. Similar to the LS-AAB system, the highest yield was observed in the second cycle, with later cycles facing challenges due to formation of BC and polysaccharide gel. On average, the LSK-AAB system produced more BC (37.0 ± 3.0 mg/L) than did the LS-AAB system, a result of the higher biomass present in the LSK-AAB setup (Table 2). At the same time, dissolved oxygen levels were significantly reduced (averaging 2.53 ± 0.10 ppm). These results underscore the delicate balance between maximizing acetic acid production and managing inhibitory by-products.

Visualization of bacterial cellulose biofilm formation during acetification with SEM

SEM was employed to investigate the adsorption of A. pasteurianus UMCC 2951 cells in the LS-AAB and LSK-AAB systems. In the LS-AAB system, extensive cell adsorption was evident as early as one day after inoculation, seen as scattered rod shapes (Fig. 3A). As acetification progressed, the cell density significantly increased, peaking at the end of the first operational phase (Fig. 3B). At that time, extracellular secretion of cellulose microfibrils became evident (square frame in Fig. 3B), leading to the formation of a distinct fibrillar network around the cells. This observation confirmed the production of BC microfibrils by A. pasteurianus UMCC 2951.

Figure 3 SEM images illustrating A. pasteurianus.

UMCC 2951 adherence and BC formation during acetification in LS-AAB and LSK-AAB systems. (A) Cells adsorbed on LS at one day post-inoculation; (B) BC secretion from LS-AAB (square frame) at the end of the first operational phase; (C) BC secretion from LSK-AAB and biofilm formation (square frame) by the end of the third operational phase; (D) Complete encapsulation of LSK by BC at the end of the fifth operational phase (square frame).

For the LSK-AAB system, peak density of adhered cells was achieved at the end of the third operational phase (Fig. 3C). As the cellular biomass was greater compared to the LS-AAB system, BC production was also greater and its accumulation more pronounced. By the fifth operational phase, the LSK-AAB system was completely encapsulated by BC (square frame in Fig. 3D).

Discussion

This study aimed to accelerate acetification by utilizing luffa sponge (LS) as a substrate for acetic acid bacteria (AAB), and by further coating the LS with κ-carrageenan (LSK). Cell quantity in the LS-AAB and LSK-AAB systems was noticeably higher than in the planktonic process; this is due to LS and LSK assisting in cell retention and proliferation (Krusong et al., 2020). In addition, the κ-carrageenan coating was found to significantly enhance the acetification process, achieving the highest rate of 3.33 ± 0.05 g/L d. The LSK-AAB system also demonstrated a shorter startup phase of 7 days and completed five operational cycles within 30 days, comprehensively outperforming both the planktonic cell and LS-AAB systems. This advancement is particularly noteworthy as previous studies, such as Krusong & Tantratian (2014), did not explore the use of κ-carrageenan in acetification.

The detailed analysis of L. cylindrica LS physical and chemical characteristics also adds value to the literature. Notably, the bulk density and porosity surpass those reported for L. aegyptiaca by Ogunsina, Adegbenjo & Opeyemi (2010), indicating potential for broader application of L. cylindrica LS in biotechnological processes. Meanwhile, the observed resilience of A. pasteurianus UMCC 2951 under high acidity aligns with the findings of Song et al. (2022) and reinforces the robustness of this bacterial strain in challenging environments.

A major strength of this study is its comprehensive exploration of acetification systems, including planktonic cells, LS-AAB, and LSK-AAB, which provides a holistic understanding of the acetification process. In addition, the observation of bacterial cellulose biofilm formation by SEM offers further valuable insights into the interaction between AAB cells and the LS matrix.

However, the study also encountered limitations. The formation of BC and polysaccharide gel in later cycles of the LS-AAB and LSK-AAB systems presented barriers to oxygen diffusion (Fernandes et al., 2020), with consequent reduction of acetification efficiency. In the planktonic process, where there was no BC formation, oxygen transfer remained efficient; however, the low microbial population resulted in a low acid yield (Román-Camacho et al., 2023). Although proactive measures were taken to remove the BC and polysaccharide gel, they consistently reemerged, an unexpected outcome that raises questions about the long-term sustainability and operational efficiency of the LS-AAB and LSK-AAB systems. Production of these compounds is indicative of a natural defensive response by A. pasteurianus, as suggested by Yun, Kim & Lee (2019), and further investigation of this phenomenon remains needed in order to optimize the acetification process.

Conclusions

The results of this study indicate that use of reciprocal shaking during submerged semi-continuous acetification can significantly enhance acetic acid production. Furthermore, this study capitalized on the inherent properties of LS, such as its high total porosity and fibrous nature, which significantly increased both biomass yield and acetic acid production, especially when modified with κ-carrageenan. These improvements were facilitated by ionic interactions between A. pasteurianus UMCC 2951 and the modified LS surface. Notably, while this study obtained promising results, the formation of BC in LS-AAB and LSK-AAB systems poses a significant operational challenge by potentially hindering mass transfer during the acetification process. Future research should focus on optimizing conditions to minimize BC formation or to manage its impact more effectively. Techniques such as varying the concentration of κ-carrageenan, exploring alternative coating materials, or modifying the physical structure of LS might provide avenues for controlling or leveraging BC production to benefit the process rather than impede it.

Additionally, further studies are warranted to explore the dual production of acetic acid and BC, particularly in specialized food products. Exploiting BC as a value-added product could open new avenues for the utilization of this system in diverse industries. Likewise, continued research into using the reciprocal shaking acetification process with a broader range of microbial strains and environmental conditions will elucidate the generalizability and robustness of the current findings. Pursuing these future directions will not only allow refinement of existing acetification methods but also facilitate enhancement of the commercial viability and sustainability of vinegar production through biotechnological innovations. Such advancements could pave the way for new product development and potentially revolutionize aspects of food processing and biomanufacturing to align with sustainable resource use and market demands.

Supplemental Information

Supplemental Information 1 Raw data for preparation of the information presented in Tale 1

Characteristics of LS as an adsorbent forA. pasteurianus UMCC 2951 in the semi-continuous acetification process

Supplemental Information 2 Raw data for preparation of the information presented in Tale 2

Comparative analysis of acetification processes using planktonic cells, LS-AAB, and LSK-AAB.

Supplemental Information 3 Raw data for preparation of the information presented in Fig. 2: Acetification byA. pasteurianus UMCC 2951 in a reciprocating shaking system at 32 ± 2°C

(A) Planktonic cells. (B) Cells adsorbed on luffa sponge (LS). (C) Cells adsorbed on luffa sponge coated κ-carrageenan (LSK).

Special gratitude was also due to Prof. Dr. Anthony Keith Thompson, whose expert editing and insightful contributions significantly enhanced the quality of the manuscript.

Abbreviations

AAB acetic acid bacteria

ANOVA one-way analysis of variance

BC bacterial cellulose

CDW cell dry weight

EPS extracellular polymeric substance

LS luffa sponge

LS-AAB luffa sponge-acetic acid bacteria

LSK luffa sponge coated with κ-carrageenan

LSK-AAB luffa sponge coated with κ-carrageenan-acetic acid bacteria

QAP quick acetification process

SEM scanning electron microscopy

Additional Information and Declarations

Competing Interests

Author Contributions

Data Availability

The authors declare there are no competing interests.

Wiramsri Sriphochanart performed the experiments, analyzed the data, prepared figures and/or tables, authored or reviewed drafts of the article, and approved the final draft.

Warawut Krusong conceived and designed the experiments, analyzed the data, authored or reviewed drafts of the article, and approved the final draft.

Nialmas Samuela performed the experiments, prepared figures and/or tables, authored or reviewed drafts of the article, and approved the final draft.

Pichayada Somboon analyzed the data, authored or reviewed drafts of the article, and approved the final draft.

Panmanas Sirisomboon analyzed the data, authored or reviewed drafts of the article, and approved the final draft.

Jiraporn Onmankhong analyzed the data, authored or reviewed drafts of the article, and approved the final draft.

Soisuda Pornpukdeewattana analyzed the data, authored or reviewed drafts of the article, and approved the final draft.

Theppanya Charoenrat conceived and designed the experiments, analyzed the data, prepared figures and/or tables, authored or reviewed drafts of the article, and approved the final draft.

The following information was supplied regarding data availability:

The raw data used for illustrating Fig. 2 and for constructing Tables 1 and 2, and statistical analysis, are available in the Supplementary Files.

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
