# Peer review of "Enhancing small-scale acetification processes using adsorbed Acetobacter pasteurianus UMCC 2951 on κ-carrageenan-coated luffa sponge"

_PeerJ, doi:10.7717/peerj.17650_

## Round 0.1 · original submission · Major Revisions

Please, authors, you can see reviewers have found some merit to your work, but also raised a lot of concerns.

Kindly address them to the very best detail. Please, pay attention to the reviewer who attached an annotated manuscript to this review, and make sure to provide responses where appropriate in your reply to concerns raised.

Make an effort to update your references.

Look forward to your revised manuscript.

Reviewer 1 ·

Basic reporting

Although a revision of the English grammar is suggested, the paper is well-structured and results are presented clearly.
Literature is outdated. No papers after 2021 are present.

Experimental design

No comment.

Validity of the findings

No comment.

Annotated reviews are not available for download in order to protect the identity of reviewers who chose to remain anonymous.

Reviewer 2 ·

Basic reporting

The work is significant. Find below my comments to improve the manuscript.
Line 25
Provide the research gap and the novelty statement.
The materials used and the details and conditions of experimental procedures have to be described with sufficient clarity, thus allowing qualified operators to repeat the work . Provide the brand and manucfacturers name of the equipments. Also, provide the SEM conditions used. I can see a lot in the materials and methods section. Correct them. Please indicate the city and country for the chemicals and instruments used once they have been given.
Line 235
Provide the GC-FID conditions used. Provide the chromatogram in the supplementary figure.
Line 250
Please clarify whether duplicate or triplicate analyses were carried out for analysis.
Results and discussion
Statistical analysis must also be discussed comprehensively in the results and discussion section.
The paper's discussion is not appropriate and needs to be modified with some recent references and some excessive explanations removed.
The authors must focus on discussing and comparing their findings with previous reports on this field. Overall, a comparison with existing literature is lacking and should be incorporated. A comprehensive discussion should include the results, the trend(s), the reasons for the trend(s) obtained, and a comparison with other studies. The current form of results and discussion is more like general reporting.
Conclusion.
Also, highlight more about future studies that need to be done.
Figure 3
Use red arrows to pinpoint what you are implying.
The manuscript article requires grammar, sentence structure, and reference format revision. Overused Passive voice in the manuscript seems hard to read. Please try to reword the phrases in the active voice. Grammar and punctuation mistakes. For consistency, please use the manuscript in just one English style (a non-variant British or British style, American style, etc.). There are phrases with the verb in the wrong tense. Sentences with words misspelled. Words are overused or unnecessary.

Experimental design

all comments are above

Validity of the findings

all comments are above

Additional comments

The work is significant. Find below my comments to improve the manuscript.
Line 25
Provide the research gap and the novelty statement.
The materials used and the details and conditions of experimental procedures have to be described with sufficient clarity, thus allowing qualified operators to repeat the work . Provide the brand and manucfacturers name of the equipments. Also, provide the SEM conditions used. I can see a lot in the materials and methods section. Correct them. Please indicate the city and country for the chemicals and instruments used once they have been given.
Line 235
Provide the GC-FID conditions used. Provide the chromatogram in the supplementary figure.
Line 250
Please clarify whether duplicate or triplicate analyses were carried out for analysis.
Results and discussion
Statistical analysis must also be discussed comprehensively in the results and discussion section.
The paper's discussion is not appropriate and needs to be modified with some recent references and some excessive explanations removed.
The authors must focus on discussing and comparing their findings with previous reports on this field. Overall, a comparison with existing literature is lacking and should be incorporated. A comprehensive discussion should include the results, the trend(s), the reasons for the trend(s) obtained, and a comparison with other studies. The current form of results and discussion is more like general reporting.
Conclusion.
Also, highlight more about future studies that need to be done.
Figure 3
Use red arrows to pinpoint what you are implying.
The manuscript article requires grammar, sentence structure, and reference format revision. Overused Passive voice in the manuscript seems hard to read. Please try to reword the phrases in the active voice. Grammar and punctuation mistakes. For consistency, please use the manuscript in just one English style (a non-variant British or British style, American style, etc.). There are phrases with the verb in the wrong tense. Sentences with words misspelled. Words are overused or unnecessary.

---

## Round 0.2 · accepted · Accept

I confirm the revised manuscript addressed all concerns raised by reviewers. I am very satisfied with the revised manuscript, and approve it for publication. Thank you authors for finding PeerJ as your journal of choice. Looking forward to your future scholarly contributions. Congratulations

Reviewer 1 ·

Basic reporting

no comment

Experimental design

no comment.

Validity of the findings

no comment

Additional comments

Autohrs have made an outstandig effort to solve all the issues. Great work.

Reviewer 2 ·

Basic reporting

ok

Experimental design

ok

Validity of the findings

ok

Additional comments

Accept